

# Assessing Spacing Impact on Coherent Features in a Wind Turbine Array Boundary Layer

Naseem Ali, Nicholas Hamilton, and Raúl Bayoán Cal

*Department of Mechanical and Materials Engineering,*

*Portland State University, Portland, OR 97207*





# Abstract

As wind farms become larger, the spacing between turbines becomes a significant design element that imposes serious economic constraints. Effects of turbine spacing on the power produced and flow structure are crucial for future development of wind energy. To investigate the turbulent flow structures in a $4 \times 3$ Cartesian wind turbine array, a wind tunnel experiment was carried out parameterizing the streamwise and spanwise wind turbine spacing. Four cases were chosen spacing turbines by 6 diameters ($D$) or $3D$ in the streamwise, and $3D$ or $1.5D$ in the spanwise direction. Data were obtained experimentally using stereo particle-image velocimetry. Mean streamwise velocity showed maximum values upstream of the turbine with the spacing of $6D$ and $3D$, in the streamwise and spanwise direction, respectively. Fixing the spanwise turbine spacing to $3D$, variations in the streamwise spacing influence the turbulent flow structure and the power available to following wind turbines. Quantitative comparisons were made through spatial averaging, shifting measurement data and interpolating to account for the full range between devices to obtain data independent of array spacing. The largest averaged Reynolds stress is seen in cases with spacing of $3D$ and $3D$, in the streamwise and spanwise direction, respectively. Snapshot proper orthogonal decomposition was employed to identify the flow structures based on the turbulence kinetic energy content. The maximum turbulence kinetic energy content in the first POD mode compared with other cases is seen for turbine spacing of $6D \times 1.5D$. The flow upstream of each wind turbine converges faster than the flow downstream according to accumulation of turbulence kinetic energy by POD modes, regardless of spacing. The streamwise-averaged profile of the Reynolds stress is reconstructed using a specific number of modes for each case; the case of $6D \times 1.5D$ spacing shows the fastest reconstruction. Intermediate modes are also used to reconstruct the averaged profile and show that the intermediate scales are responsible for features seen in the original profile. The variation in streamwise and spanwis spacing leads to changing the background structure of the turbulence, where the color map based on barycentric map and anisotropy stress tensor provides a new perspective on the nature of the perturbations within the wind turbine array. The impact of the streamwise and spanwise spacings on power produced is quantified, where the maximum production corresponds with the case of greatest turbine spacing.

6  PACS numbers:





## I. INTRODUCTION

Allowing insufficient space between wind turbines in an array leads to decreased performance through wake interaction, decreased wind velocity and an increased in the accumulated fatigue loads on downstream turbines. Wind turbine wakes lead to an average loss of 10-20% of the total potential power output of wind turbine array (Barthelmie et al. 2007). Extensive experimental and numerical studies focus on wake properties in terms of the mean flow characteristics used to obtain estimates of power production (Chamorro and Porté-Agel 2009, Cal et al. 2010, Calaf et al. 2010, Chamorro and Porté-Agel 2011). Wake growth depends on the shape and magnitude of the velocity deficit, which is in turn in a function relying on the surface roughness, flow above the canopy and spacing between the turbines.

Although there are many studies dealing with the effect of the density of turbines on the wake recovery, it is still a debated question. The actual spacing of wind turbines can vary greatly from one array to another. For example, in the Nysted farm, spacing is 10.5 diameters ($D$) downstream by $5.8D$ spanwise at the exact row (ER). The wind direction at the ER is 278° and mean wind direction can slightly offset from ER by $\pm$ 15° Barthelmie et al. (2010). In the Horns Rev farm, spacing between devices is $7D$, although aligned with the bulk flow direction spacing is as much as $10.4D$. Barthelmie and Jensen (2010) showed that the spacing in the Nysted farm is responsible for 68-76% of the farm efficiency variation. Hansen et al. (2012) pointed out that variations in the power deficit are almost negligible when spacing is approximately $10D$ at the Horns Rev farm, in contrast to limited spacings that present a considerable power deficit. González-Longatt et al. (2012) found that when the streamwise and spanwise spacing increased, the wake coefficient, which represents the ratio of total power output with and without wake effects, is increased. Further, the effect of the incoming flow direction on the wake coefficient increased when the spacing of the array is reduced. Meyers and Meneveau (2012) studied the optimal spacing in a fully developed wind farm under neutral stratification and flat terrain. The results highlighted that, depending on the ratio of land and turbine costs, the optimal spacing might be $15D$ instead of $7D$. Stevens (2015) pronounced that the optimal spacing depends on the length of the wind farm in addition to the factors suggested in Meyers and Meneveau (2012). Nilsson et al. (2015) performed large eddy simulations (LES) of the Lillgrund wind farm, where pre-generated

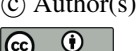



turbulence and wind shear were imposed in the computational domain to simulate realistic atmospheric conditions. In the Lillgrund wind farm, the actual spacing is $3.3D$ and $4.6D$ in the streamwise and spanwise directions. A turbine is missing near to the center of the wind farm, demonstrating the effects of a farm with limited spacing and one with sufficient spacing in otherwise identical operating conditions. The results of Nilsson et al. (2015) are highly applicable in the current study, although their foci are on turbulence intensity effects and yaw angle.

Further investigations in array optimization have been undertaken by changing the alignment of the wind farm, often referred to as staggered wind farms. Meyers and Meneveau Meyers and Meneveau (2010) compared aligned versus staggered wind farms; the latter yielding a 5% increase in extracted power. Yang et al. (2012) used LES to study the influence of the streamwise and spanwise spacing on the power output in aligned wind farms under fully developed regime. Their work confirmed that power produced by the turbines scales with streamwise spacing more than with the spanwise spacing. Wu and Porté-Agel (2013) investigated turbulent flow within and above aligned and staggered wind farms under neutral conditions using LES. Cumulative wakes are shown to be subject to strong lateral interaction in the staggered case. In contrast, lateral interaction between cumulative wakes is negligible in the aligned wind farm. Archer et al. (2013) quantified the influence of wind farm layout on the power production, verifying that increasing the turbine spacing in the predominant wind direction maximized the power production, regardless of device arrangement in the wind farm. Stevens et al. (2016) used LES model to investigate the power output and wake effects in aligned and staggered wind farms with different streamwise and spanwise turbine spacings. In the staggered configuration, power output in a fully developed flow depends mainly on the spanwise and streamwise spacings, whereas in the aligned configuration, power strongly depends on the streamwise spacing.

As wind farms become larger, the land costs and availability represent critical factors in the overall value of the wind farm. Spacing between the turbines is an important design factor in terms of overall wind farm performance and economic constraints. Investigation of wind farms with limited spacing is important in order to quantify the effects of wind turbine wake interaction on the power production. The current work compares the turbulent flow in various configurations of the array, where the streamwise and spanwise spacings are varied. The performance of the arrays is characterized by analyzing the mean velocity,




Reynolds shear stress, flux of mean kinetic energy, and power production. Proper orthogonal
decomposition (POD) is employed to identify coherent structures of the turbulent wake
associated with variations in spacing. The Reynolds stresses are reconstructed from POD
basis, demonstrating variation in rates of convergence according to wind turbine spacing.
Finally the anisotropy stress tensor is discussed to quantify the structure of the stress tensor
based on the invariant for the various spacings.

## II. THEORY

### A. Snapshot Proper Orthogonal Decomposition

POD is a mathematical tool that derives optimal basis functions from a set of measure-
ments, decomposing the flow into modes that express the most dominant features. The
technique, which was presented in the frame of turbulence by Lumley (1967), categorizes
structures within the turbulent flow depending on their energy content. Sirovich (1987)
presented the snapshot POD, that relaxes the computational difficulties of the classical or-
thogonal decomposition. POD has been used to describe coherent structures for different
flows, such as axisymmetric mixing layer (Glauser and George 1987), channel flow (Moin
and Moser 1989), atmospheric boundary layer (Shah and Bou-Zeid 2014), wake behind disk
(Tutkun et al. 2008), subsonic jet (Tirunagari et al. 2012) and wind turbine wake flow (An-
dersen et al. 2013, Hamilton et al. 2015, Bastine et al. 2014, VerHulst and Meneveau 2014,
Ali et al. 2016).
The flow field, taken as the fluctuating velocity, can be represented as $u = u(\vec{x}, t^n)$, where
$\vec{x}$ and $t^n$ refer to the spatial coordinates and time at sample $n$, respectively. A set of the
orthonormal basis functions, $\phi$, can be presented as

$$\phi = \sum_{n=1}^{N} A(t^n) u(\vec{x}, t^n). \tag{1}$$

The largest projection can be determined using the two point correlation tensor and Fred-
holm integral equation

$$\int_\Omega \frac{1}{N} \sum_{n=1}^{N} u(\vec{x}, t^n) u^T(\vec{x}', t^n) \phi(x') dx' = \lambda \phi(x), \tag{2}$$





where $R(\vec{x}, \vec{x}')$ is a spatial correlation between two points $\vec{x}$ and $\vec{x}'$, $N$ is the number of
snapshots, $T$ signifies the transpose of a matrix, and $\lambda$ are the eigenvalues. To acquire the
optimal basis functions, the problem is reduced to an eigenvalue decomposition denoted as
$[C][G] = \lambda[G]$, where $C$, $G$ and $\lambda$ are the correlation tensor, basis of eigenvectors, and eigen-
values, respectively. The POD eigenvectors illustrate the spatial structure of the turbulent
flow and the eigenvalues measure the energy associated with corresponding eigenvectors.
The summation of the eigenvalues presents the total turbulent kinetic energy ($E$) in the
flow domain. The cumulative kinetic energy fraction $\eta$ and the normalized energy content
of each mode $\xi$ can be represented as $\eta_n = \frac{\sum_{n=1}^{n} \lambda_n}{\sum_{n=1}^{N} \lambda_n}$ and $\xi_n = \frac{\lambda_n}{\sum_{n=1}^{N} \lambda_n}$. POD is particularly
useful in rebuilding the Reynolds shear stress using a limited set ($N_{lm}$) of eigenfunctions as
follows,

$$\langle u_i u_j \rangle = \sum_{n=1}^{N_{lm}} \lambda_n \phi_i^n \phi_j^n. \tag{3}$$

### B.   Anisotropy Stress Tensor
Turbulence is often described through the Reynolds stress tensor. Rotta (1951) developed
the Reynolds stress anisotropy tensor, as $a_{ij} = \overline{u_i' u_j'} - \frac{2}{3} k \delta_{ij}$, where $\delta_{ij}$ is the Kronecker
delta and $k$ represents the turbulent kinetic energy.  The deviatoric tensor is obtained,
$b_{ij} = \overline{u_i' u_j'} / \overline{u_k' u_k'} - \frac{1}{3} \delta_{ij}$. The second and third scalar invariants are defined as $6\eta^2 = b_{ij} b_{ji}$
and $6\xi^3 = b_{ij} b_{jk} b_{ki}$, respectively (see Pope (2000), Lumley and Newman (1977) for more
details).  The second invariant, $\eta$, measures the degree of the anisotropy and the third
invariant, $\xi$, specifies the state of turbulence. Alternatively, the eigenvalue decomposition
of the normalized Reynolds stress anisotropy tensor can be used to derive the the second
and third invariants as $\eta^2 = \frac{1}{3}(\lambda_1^2 + \lambda_1 \lambda_2 + \lambda_2^2)$ and $\xi^3 = -\frac{1}{2} \lambda_1 \lambda_2 (\lambda_1 + \lambda_2)$. In an attempt
to additional promote the study of turbulence anisotropy, Banerjee et al. (2007) presented
a linearized anisotropy tensor invariants, termed barycentric map (BM) as follows,

$$\hat{b}_{ij} = C_{1c} \begin{pmatrix} 2/3 & 0 & 0 \\ 0 & -1/3 & 0 \\ 0 & 0 & -1/3 \end{pmatrix} + C_{2c} \begin{pmatrix} 1/6 & 0 & 0 \\ 0 & 1/6 & 0 \\ 0 & 0 & -1/3 \end{pmatrix} + C_{3c} \begin{pmatrix} 0 & 0 & 0 \\ 0 & 0 & 0 \\ 0 & 0 & 0 \end{pmatrix}, \tag{4}$$

where $C_{1c}$, $C_{2c}$ and $C_{3c}$ are the coefficients that represent the boundary of the barycentric



TABLE I: Summary of the special turbulence cases described by the barycentric map.

| Cases | Eigenvalues |
|---|---|
| Three-component | $\lambda_1 = \lambda_2 = \lambda_3 = 0$ |
| Two-component | $\lambda_1 = \lambda_2 = \frac{1}{6}, \lambda_3 = -\frac{1}{3}$ |
| One-component | $\lambda_1 = \frac{2}{3}, \lambda_2 = \lambda_3 = -\frac{1}{3}$ |

map. The BM coefficients are determined as $C_{1c} = \lambda_1 - \lambda_2$, $C_{2c} = 2(\lambda_2 - \lambda_3)$, and $C_{3c} = 3\lambda_3 + 1$. The three basis matrices in equation (4) represent the three vertices of the equilateral triangle, with the following coordinates $(x_{1c}, y_{1c})$, $(x_{2c}, y_{2c})$ and $(x_{3c}, y_{3c})$. Table I presents the three turbulence states corresponding with the vertices of the BM, which also correspond to either isotropic (three-component), one- or two-component turbulence. As a result, any realizable turbulence state can be represented as follows,

$$x_{new} = C_{1c}x_{1c} + C_{2c}x_{2c} + C_{3c}x_{3c}, \tag{5}$$

$$y_{new} = C_{1c}y_{1c} + C_{2c}y_{2c} + C_{3c}y_{3c}. \tag{6}$$

Emory and Iaccarino (2014) also introduced a color map based visualization technique that aids to interpret the spatial distribution of the normalized anisotropy tensor. In this case, they attributed to each vertex of the barycentric map an RGB (Red-Green-Blue) color, see figure 1 for more details. This color map technique combines the coefficients $C_{1c}$, $C_{2c}$ and $C_{3c}$ to generate an RGB map such that,

$$\begin{bmatrix} R \\ G \\ B \end{bmatrix} = C_{1c}^* \begin{bmatrix} 1 \\ 0 \\ 0 \end{bmatrix} + C_{2c}^* \begin{bmatrix} 0 \\ 1 \\ 0 \end{bmatrix} + C_{3c}^* \begin{bmatrix} 0 \\ 0 \\ 1 \end{bmatrix}. \tag{7}$$

where $C_{ic}^*$ are the modified coefficients that can be determined as $C_{ic}^* = (C_{ic}^* + 5)^{0.65}$. As a result, one-component turbulence is associated to the red color, two-component turbulence to green, and three-component (isotropic turbulence) to blue, see figure 1. Representing the anisotropy tensor with both techniques, the Lumley and barycentric maps, provides a better visual understanding of the anisotropy of turbulent flows, especially for large data sets. The




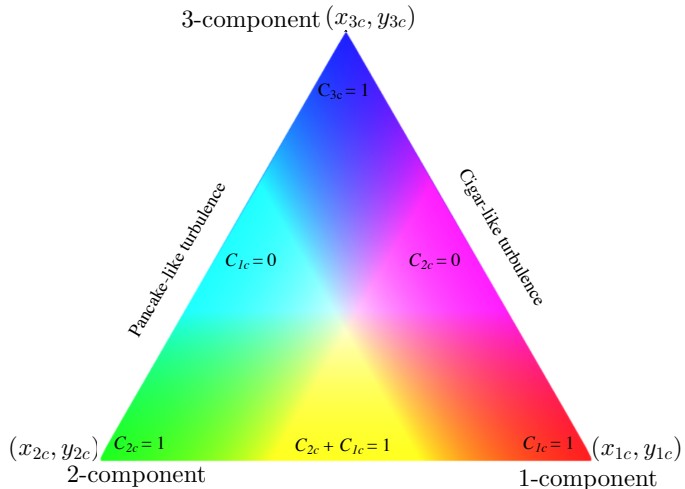

FIG. 1: Schematic representation of the Barycentric map (BM) with color map.

anisotropy invariant map has been used to examine different types of flows, including pipe
and duct flows (Antonia et al. 1991, Krogstad and Torbergsen 2000) as well as the wake of
a wind turbine (Gómez-Elvira et al. 2005, Hamilton and Cal 2015). Here we will used the
anisotropy stress tensor is employed to quantify the effect of the spacing on the turbulence
states.

## III.   EXPERIMENTAL DESIGN

A $4 \times 3$ array of wind turbines was placed in the closed- circuit wind tunnel at Portland
State University to study the effects due to variation in streamwise and spanwise spacing
in a wind turbine array. The dimensions of the wind tunnel test section are 5 m (long), 1.2
m (wide) and 0.8 m (high). The entrance of the test section is conditioned by the passive
grid, which consists of 7 horizontal and 6 vertical rods, to introduce large-scale turbulence.
Nine vertical acrylic strakes, located at 0.25 m downstream of the passive grid and 2.15 m
upstream of the first row of the wind turbine, were used to modify the inflow. The thickness
of the strakes is 0.0125 m and are spaced every 0.136 m across the test section. Surface
roughness was introduced to the wall as a series of chains with a diameter of 0.0075 m,
spaced 0.11 m apart. Figure 2 shows the schematic of the experimental setup.



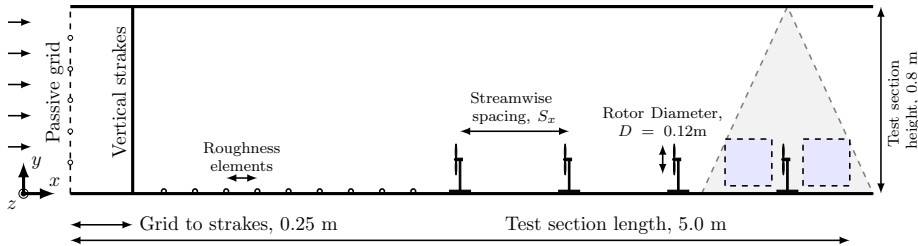

FIG. 2: Experimental Setup. Dashed gray lines indicate the placement of the laser sheet relative to the model wind turbine array. Filled gray boxes indicate measurement locations discussed below.

Sheet steel of 0.0005 m thick was used to construct the 3-bladed wind turbine rotors. The diameter of the rotor was $D = 0.12$ m, equal to the height of the turbine tower. Each rotor blade was pitched at 15° out of a plane at the root and 5° at the tip. These angles were chosen to provide angular velocity that correlates with required ranges of tip-speed ratio. A DC electrical motor of 0.0013 m diameter and 0.0312 m long formed the nacelle of the turbine and was aligned with the flow direction. A torque-sensing system was connected to the DC motor shaft following the design outlined in Kang and Meneveau (2010). The torque sensor consists of a strain gauge, Wheatstone bridge and the Data Acquisition with measuring software to collect the data.

The flow field was sampled under neutral stratification in four configurations of a model-scale wind turbine array, classified as $C_{S_x \times S_z}$, shown in Table II. Permutations of the streamwise spacing ($S_x$) of $6D$ and $3D$ and spanwise spacing ($S_z$) of $3D$ and $1.5D$ are examined. Thus, the four cases present aligned wind farm; the staggered wind farm does not consider in this study. Stereoscopic particle image velocimetry (SPIV) was used to measure streamwise, wall-normal and spanwise instantaneous velocity at the upstream and downstream of the wind turbine at the center line of the fourth row as shown in figure 3. At each measurement location, 2000 images were taken, to ensure convergence of second-order statistics. SPIV equipment is LaVision and consists of a Nd:Yag (532nm, 1200mJ, 4ns duration) double-pulsed laser and four 4 MP ImagerProX CCD cameras positioned for the upstream and downstream of the wind turbine. Neutrally buoyant fluid particles of diethyl hexyl sebacate were introduced to the flow and allowed to mix. Consistent seeding density was maintained in order to mitigate measurement errors. The laser sheet was approximately 0.001 m thick with less than 5 mrad divergence angle. Each measurement window was 0.2





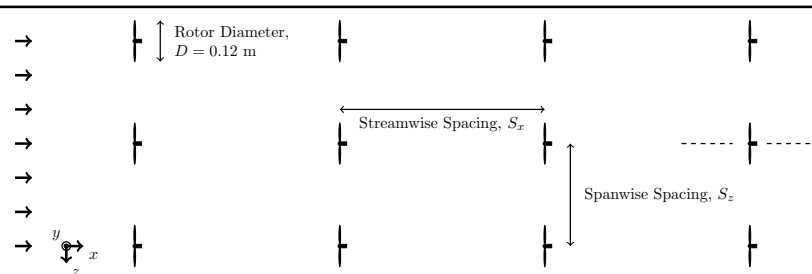

FIG. 3: Top view of 4 by 3 wind turbine array. The dash lines at the last row centerline turbine represent the measurement locations.

m × 0.2 m aligned with the center of each turbine, parallel to the bulk flow. A multi-pass
fast Fourier transformation was used to process the raw data into vector fields. Erroneous
measurement of the vector fields were replaced using Gaussian interpolation of neighboring
vectors.

TABLE II: Streamwise and spanwise spacing of the experimental tests.

| Cases | $S_x$ | $S_z$ | Occupied Area |
|---|---|---|---|
| $C_{6\times3}$ | $6D$ | $3D$ | $18D^2$ |
| $C_{3\times3}$ | $3D$ | $3D$ | $9D^2$ |
| $C_{3\times1.5}$ | $3D$ | $1.5D$ | $4.5D^2$ |
| $C_{6\times1.5}$ | $6D$ | $1.5D$ | $9D^2$ |

## IV. RESULTS

### A. Statistical Analysis.

Characterization of the wind turbine wake flow is presented by the streamwise mean
velocity, Reynolds shear stress, and flux of kinetic energy, with the aim to understand the
influence of turbine-to-turbine spacing. Figure 4 presents the streamwise normalized mean
velocity, $U/U_\infty$, upstream and downstream of each wind turbine for the cases $C_{6\times3}$, $C_{3\times3}$,
$C_{3\times1.5}$ and $C_{6\times1.5}$. $U_\infty$ is about $5.5 \, \text{ms}^{-1}$ and represents the inflow velocity at hub height. The
left and right contour plots of each case present the flow upstream and downstream of each




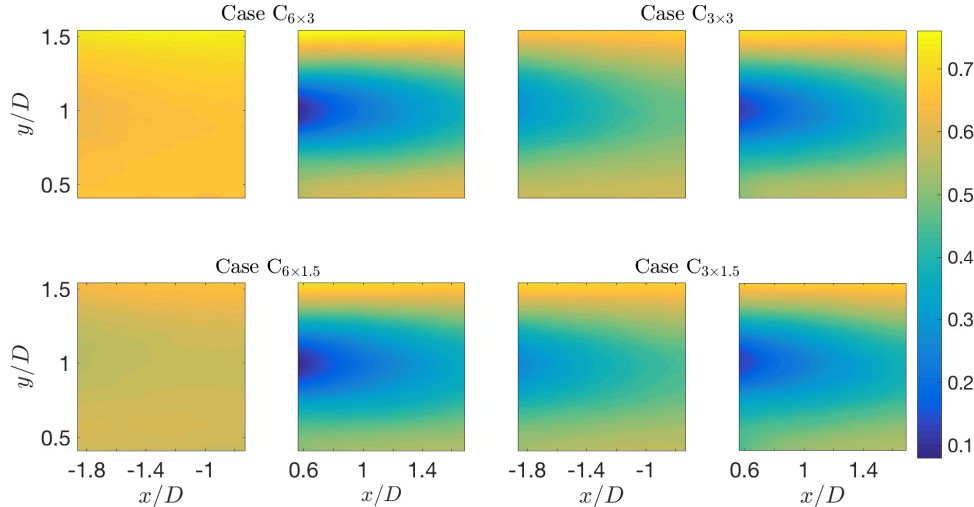

FIG. 4: Normalized streamwise velocity, $U/U_\infty$, at upstream and downstream of the cases $C_{6\times3}$, $C_{3\times3}$, $C_{3\times1.5}$, and $C_{6\times1.5}$.

turbine, respectively. At upstream measurement window, case $C_{6\times3}$ exhibits the largest
streamwise mean velocities due to greater recovery of the flow upstream of the turbine.
Although the streamwise spacing of case $C_{6\times1.5}$ is similar that of case $C_{6\times3}$, the former
shows reduced hub height velocity. The normalized mean velocity is about 0.567 compared
with 0.66 in case $C_{6\times3}$, confirming the influence of the spanwise spacing on wake evolution
and flow recovery. Variations perceived between case $C_{3\times3}$ and $C_{3\times1.5}$ are small, where
case $C_{3\times3}$ demonstrates higher velocities by approximately 2%. Downstream of the turbine,
the four cases show more relevant differences especially above the top tip and below the
bottom tip, where case $C_{6\times3}$, once again, shows the greatest velocities by approximately
20%. Case $C_{3\times3}$ also shows higher velocities below the bottom tip compared with cases
$C_{3\times1.5}$ and $C_{6\times1.5}$. The normalized mean streamwise velocity and the turbulence intensity in
Nilsson et al. (2015) showed similar compound wakes from the upstream and downstream
turbines and confirmed the current result of cases $C_{3\times3}$ and $C_{3\times1.5}$. In the study, there was
one location with an absent turbine and the flow was given extra space for recovery. The
recovered wake flow in Nilsson et al. (2015) is similar to the present cases $C_{6\times3}$ and $C_{6\times1.5}$.
Figure 5 compares the in-plane normalized Reynolds shear stress $-\overline{uv}/U_\infty^2$ for all test
cases. The fluctuating velocities in streamwise and wall-normal direction are denoted as $u$





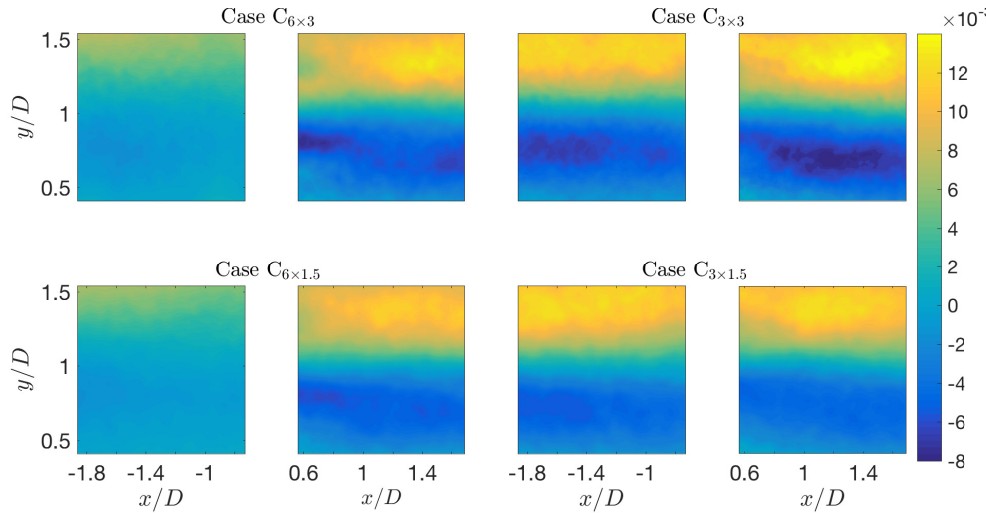

FIG. 5: Normalized Reynolds shear stress, $-\overline{uv}/U_\infty^2$, in upstream and downstream of the each measurement case.

and $v$, respectively. In the upstream window, cases $C_{3\times3}$ and $C_{3\times1.5}$ display higher stress
compared with $C_{6\times3}$ and $C_{6\times1.5}$. Although the spanwise spacing of case $C_{3\times1.5}$ is half of
case $C_{3\times3}$, no relevant differences are apparent. In the downstream window, comparison
indicates that reducing streamwise spacing increases the Reynolds shear stress. The average
value of the shear stress in the wake is 16% greater for $C_{3\times3}$ than for $C_{6\times3}$. A similar effect is
observed in case $C_{3\times1.5}$, where average value of the stress is 2% greater than that of $C_{6\times1.5}$.
The effect of spanwise spacing is more pronounced when the streamwise spacing is $3D$; the
average shear stress is approximately 20% greater in $C_{3\times1.5}$ than in $C_{3\times3}$.

### B. Averaged Profiles.

Spatial averaging of the flow statistics is undertaken by moving the upstream domain of
each case beyond its corresponding downstream domain and performing streamwise averag-
ing, following the procedure in Cal et al. (2010). Though the spatial averaging, it is possible
to compare key data from different cases taking into account the different streamwise spacing.
Streamwise averaging is denoted by $\langle\cdot\rangle_x$. Figure 6(a) shows profiles of streamwise-averaged
mean velocity for all four cases. Cases $C_{6\times3}$ and $C_{3\times1.5}$ show the largest and smallest ve-





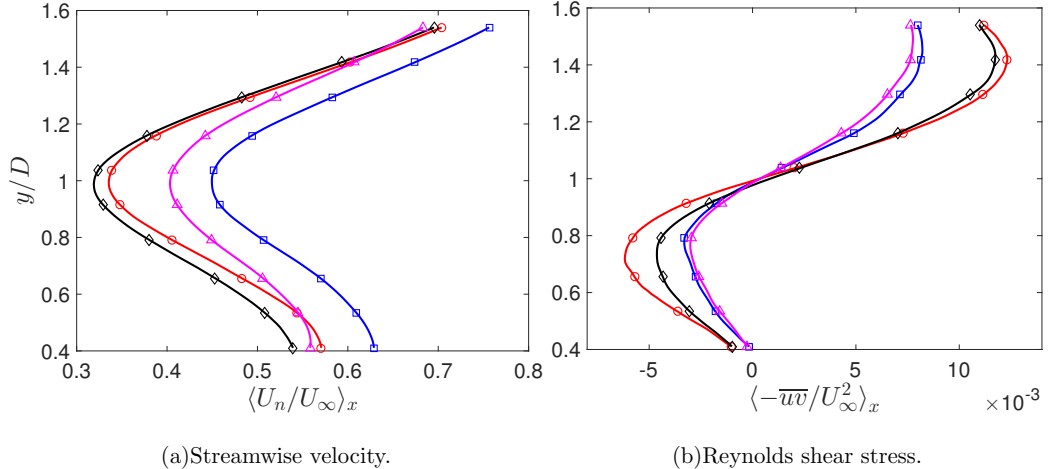

(a)Streamwise velocity.  (b)Reynolds shear stress.

FIG. 6: Streamwise-averaged profiles of streamwise velocity, and Reynolds shear stress for four different cases $C_{6\times3}$ ($\square$), $C_{3\times3}$ ($\bigcirc$), $C_{3\times1.5}$ ($\Diamond$), and $C_{6\times1.5}$ ($\triangle$).

locity deficits, respectively. At hub height, the velocity of the case $C_{6\times3}$ is approximately 2.25 ms$^{-1}$ whereas case $C_{3\times1.5}$ shows a velocity of approximately 1.6 ms$^{-1}$. Comparing to $C_{6\times3}$, the change seen in the spatially-averaged velocity is greater in $C_{3\times3}$ than in $C_{6\times1.5}$, confirming that the impact of reducing streamwise spacing is greater than changing the spanwise spacing. Interestingly, a reduction in streamwise spacing shows less effect when the spanwise spacing $S_z = 1.5D$.

Figure 6(b) contains the streamwise-averaged Reynolds shear stress $\langle -\overline{uv}/U_\infty^2\rangle_x$ for cases $C_{6\times3}$ through $C_{6\times1.5}$. Slightly decreased in $\langle -\overline{uv}/U_\infty^2\rangle_x$ are attained in case $C_{6\times1.5}$, where the spanwise spacing is reduced. Reducing spanwise spacing shows an important influence when the streamwise spacing is $x/D = 3$. The streamwise spacing plays a larger role than the spanwise spacing, *i.e.* the maximum differences between the Reynolds shear stress profiles are detected between cases $C_{6\times3}$ and $C_{3\times3}$. Interestingly, the largest difference between the spatially-averaged Reynolds shear stress is found between cases $C_{6\times3}$ and $C_{3\times3}$, located at $y/D \approx 0.7$ and $y/D \approx 1.4$. Furthermore, the four cases have approximately zero Reynolds shear stress at the inflection point located at hub height. In addition, case $C_{3\times3}$ displays the maximum Reynolds stress and case $C_{6\times1.5}$ presents the minimum stress.



### C.  Proper Orthogonal Decomposition.

Based on the velocity field, the spatially integrated turbulent kinetic energy is expressed by the eigenvalue of each POD mode. The normalized cumulative energy fraction $\eta_n$ for upstream and downstream measurement windows are presented in figure 7(a) and (b), respectively. Inset figures exhibit the normalized energy content per mode, $\xi_n$. Upstream of the turbine, cases $C_{6\times3}$ and $C_{6\times1.5}$ converge faster than cases $C_{3\times3}$ and $C_{3\times1.5}$, respectively. These results are attributed to the reduction the streamwise spacing. The convergence of case $C_{3\times3}$ is approximately coincident with case $C_{3\times1.5}$. For the downstream flow, case $C_{6\times1.5}$ converges faster than the other cases, thereafter it is ordered as $C_{6\times3}$, $C_{3\times3}$ and $C_{3\times1.5}$ in succession. The comparison between the upstream and downstream windows reveals that energy accumulates in fewer modes upstream in each case, *e.g.*, case $C_{6\times3}$ requires 14 modes to obtain 50% of the total kinetic energy in the upstream window, whereas 26 modes are required to obtain the same percentage of energy downstream of the turbine. Cases $C_{6\times1.5}$ and $C_{3\times1.5}$ show the maximum and minimum variations in $\lambda_1$, respectively. This observation can be attributed to the structure of the upstream flow of case $C_{6\times1.5}$, which is rather recovered, compared to the downstream flow, where the turbulence is high in energy content and more complex. However, the upstream and downstream windows of case $C_{3\times1.5}$ are more similar in terms of turbulence and organization. From mode 2 through 10, the starkest difference between the upstream and downstream is found in case $C_{6\times3}$. Increasing the spacing area per turbine provides room for the flow to become more homogeneous in the upstream window and exhibit the most significant momentum deficit in the wake, accounting for the differences seen in $\eta_n$ upstream and downstream.

The streamwise component of several POD modes is shown for all cases in figures 8 through 10. These modes were selected because of their importance to the flow and their variation from case to case. Figure 8 presents the first POD mode at the upstream and downstream of the considered cases. The four cases show small gradients in the streamwise direction compared to a large gradient in the wall-normal direction. Although the four cases show a divergence between the eigenvalues of the first mode, the eigenfunctions display very similar structures. For case $C_{6\times3}$ energy of the first POD mode shows decreases by 1.25% comparing the upstream eigenvalue to the downstream one, see figure 7. Smaller variations of 0.68% and 0.32% are observed in the cases $C_{3\times3}$ and $C_{3\times1.5}$, respectively. Consequently,

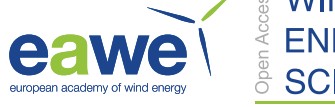



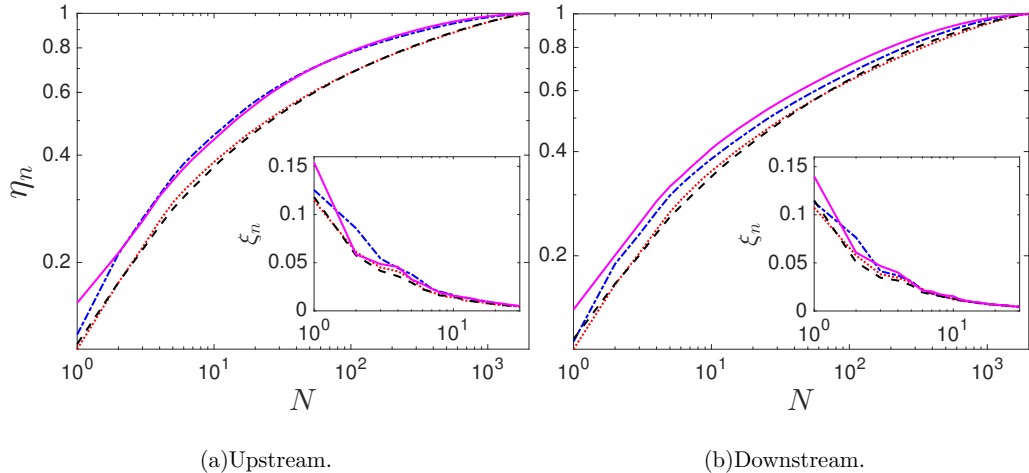

(a)Upstream.          (b)Downstream.

FIG. 7: Energy content of the POD modes for four different cases: $C_{6\times3}$ ($-\cdot-$), $C_{3\times3}$ ($\cdots$), $C_{3\times1.5}$ ($--$), and $C_{6\times1.5}$ ($-$).

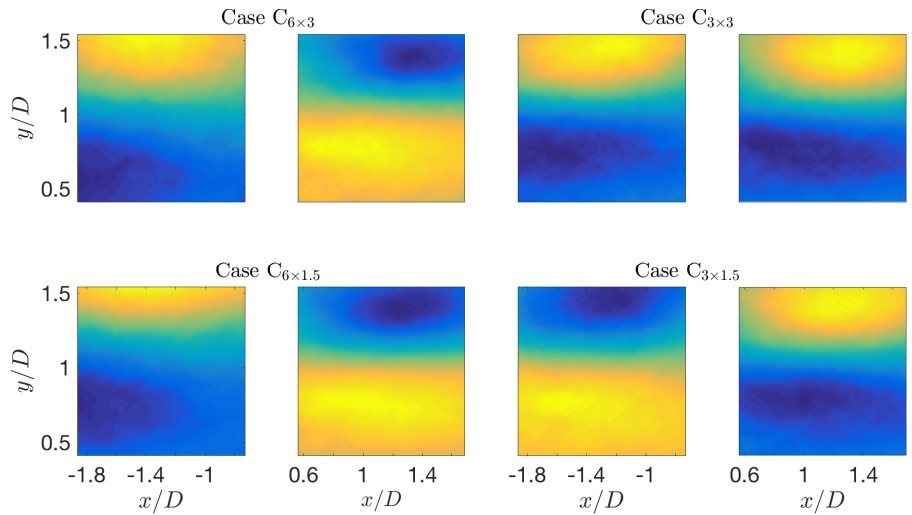

FIG. 8: The first mode upstream and downstream of the each case.

the structures of upstream and downstream of these cases are approximately equivalent. The similarity is observed between case $C_{6\times3}$ and $C_{6\times1.5}$ despite the turbulent kinetic energy difference between them about 3%.

Figure 9 presents the fifth POD mode of the four cases that show a combination of POD and Fourier (homogenous) modes in the streamwise direction. Although the fifth mode of the





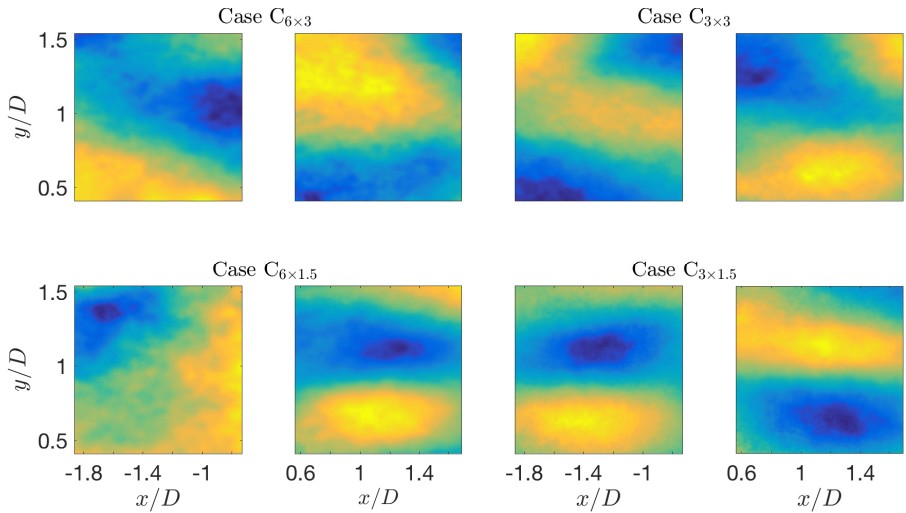

FIG. 9: The fifth mode upstream and downstream of the each case.

four cases contains ≈ 74% less energy of than the first mode, large scales are still pronounced.
Smaller features also appear in the upstream and the downstream windows. The upstream
window of cases $C_{6\times3}$, $C_{3\times3}$, and $C_{3\times1.5}$ is shifted horizontally in the downstream window.
The upstream and downstream widows of case $C_{3\times1.5}$ look like the first mode, but at a
reduced scale. The same trend is observed in the downstream window of the case $C_{6\times1.5}$.
Figure 10 presents the twentieth POD mode, where small structures become noticeable
in both upstream and downstream windows. The upstream measurement window of cases
$C_{6\times3}$ and $C_{6\times1.5}$ shows large scale structures compared with the other two cases. Although,
after mode 10, there is no significant difference in the energy content from case to case, the
structure of the modes shows a significant discrepancy between the cases confirming that
the intermediate modes associate with the inflow characterizations.
**D.    Reconstruction of Averaged Profile.**
A reduced degree of the turbulence kinetic energy is considered using only a few modes
to reconstruct the streamwise-averaged profiles of Reynolds shear stress. Reconstructions
are made using either the first mode, the first 5, 10, 25, or 50 modes to represent the
stress as shown in figure 11. Inset figures present the Reynolds shear stress construction



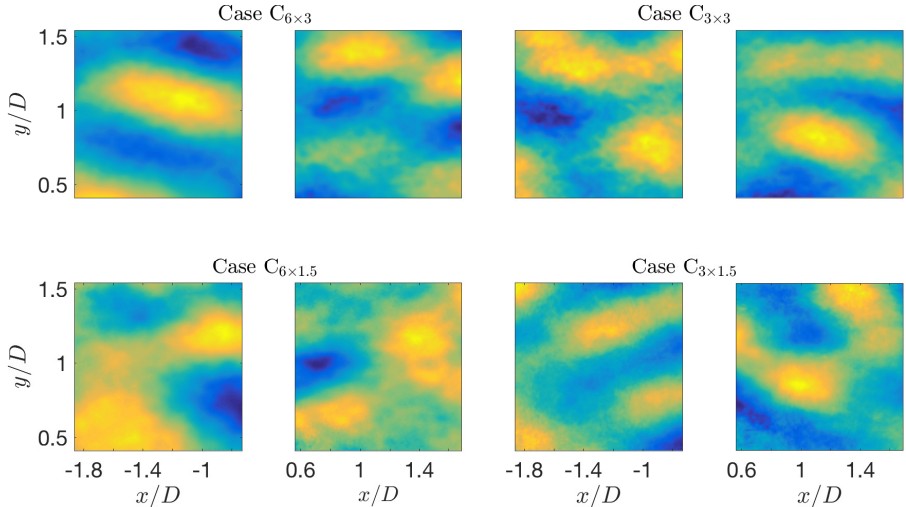

FIG. 10: The twentieth mode upstream and downstream of the each case.

using the modes 5-10, 5-25, and 5-50, respectively, excluding the first four modes isolates
contributions from intermediate modes. The black lines are the streamwise average of full
data from figure 6(b). Using an equal number of modes, case $C_{6\times1.5}$ rebuilds the profiles
of the Reynolds shear stress faster than the other cases. Case $C_{6\times3}$ also shows the fast
reconstruction and the dissimilarity with case $C_{6\times1.5}$ is mainly in the profile of first mode
(red line) and the first five modes (blue line). Cases $C_{3\times3}$ and $C_{3\times1.5}$ show approximately
the same trends in reconstruction profiles. Below hub height, the four cases show the same
trend of the first mode profiles, where the contribution in the reconstruction profiles is zero.
The first five modes display exactly the form of the full data profile of individual case. The
maximum difference between the successive reconstruction profiles occurs between the first
mode and the first five modes. The cases $C_{6\times3}$, $C_{3\times3}$ and $C_{3\times1.5}$ show moderate variation
between the profiles of the reconstructed stress resulting from first five and first ten modes
(red and green lines, respectively). After mode 10 contributions by each additional mode
are quite small, shown by pink and gray lines.
The maximum difference between the full data and the reconstructed profiles is located
at $y/D \approx 0.75$ and $y/D \approx 1.4$, where the extrema in $\langle -\overline{uv} \rangle_x$ are located. Generally, faster
reconstruction implies that the flow possesses coherent structures with a greater portion
of the total kinetic energy. Consequently, the flow characterized with greater coherence in




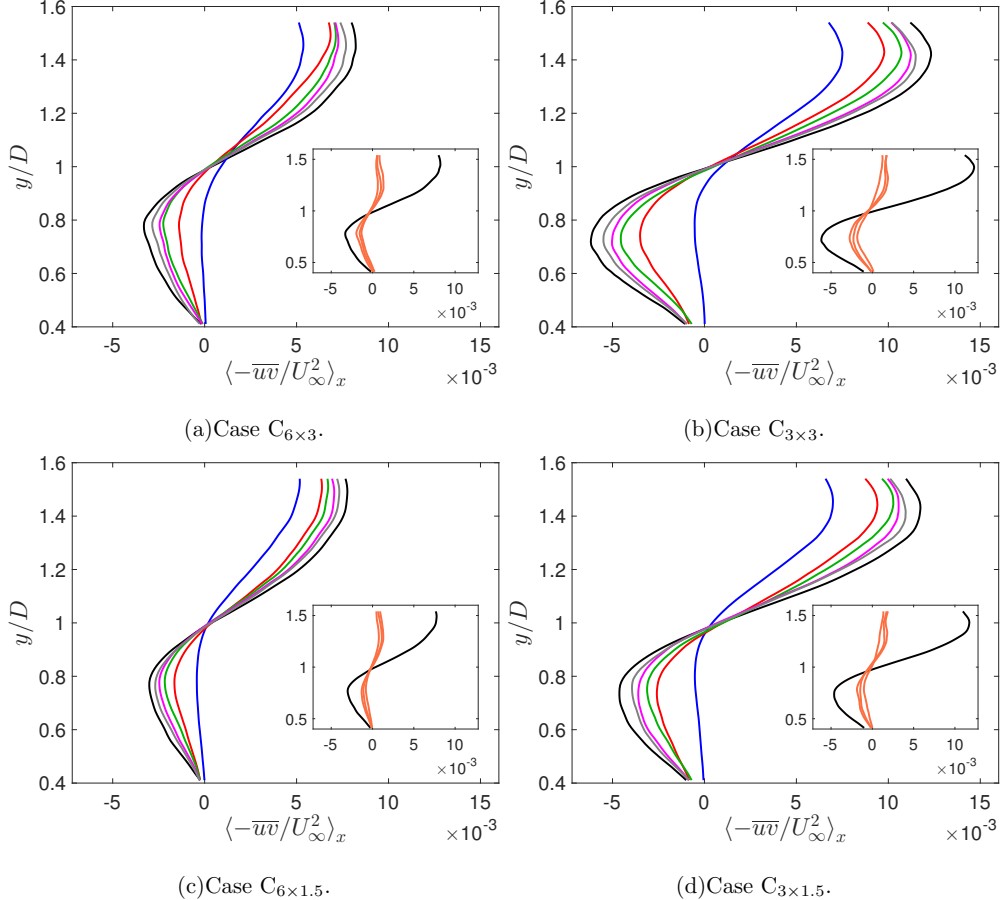

(a)Case $C_{6\times3}$.

(b)Case $C_{3\times3}$.

(c)Case $C_{6\times1.5}$.

(d)Case $C_{3\times1.5}$.

FIG. 11: Reconstruction Reynolds shear stress using: first mode (—), first 5 modes (—), first 10 modes (—), first 25 modes (—) and first 50 modes (—). Full data statistics (—). The insets show the reconstruction using modes 5-10, 5-25, and 5-50 (—).

the cases $C_{6\times3}$ and $C_{6\times1.5}$; in cases $C_{3\times3}$ and $C_{3\times1.5}$, less energetic features are observed. Thus, streamwise spacing allows for the flow to recover and therefore produce larger structures within the domain, which in comparison eclipses variations produced by the spanwise spacing.

To quantify the contribution of the moderate-scaled structures, Reynolds shear stress is reconstructed using the intermediate modes. As can be shown in the insets of figure 11, the full data profile (black line) is compared with profiles reconstructed from modes 5-10, 5-25, and 5-50 (orange lines). Surprisingly, the intermediate modes in each case approximately take the form of the full data profiles below the hub height, although the magnitudes of the





reconstructions are smaller than those of the full data statistics. Reconstruction Reynolds
shear stress in cases $C_{6\times3}$ and $C_{3\times1.5}$ show minute variations between the successive re-
construction profiles and are essentially vertical lines above the hub height. This trend is
opposite to the trend that is shown in the first mode profile. Cases $C_{3\times3}$ and $C_{3\times1.5}$ show a
difference between the successive profiles above the hub height. The maximum difference is
observed between the reconstructed profiles from modes 5-10 and from 5-25.

### E.  Anisotropy Stress Tensor

To examine the dynamics and energy transfer in the wind turbine arrays with different
streamwise and spanwise spacings, a description of the anisotropy in the upstream and
downstream of the wind turbines is presented in figure 12. A visualization of the turbulence
state is obtained via the color map representing the barycentric map as described in section
II B, where it efficiently distinguishes among the cases in terms of wake propagation and wake
interaction. The variation in the spacings changes the background turbulence structure.
The upstream of cases $C_{6\times3}$ and $C_{6\times1.5}$ shows the turbulence state close to the isotropy limit
especially in hub height region as a result of the wake recovery occurring under a relatively
long spacing distance. Below the bottom tip, these cases show pancake-like turbulence due to
the surface effect that appear deeming the perturbation of the turbines virtually negligible.
Near top tip, the flow shows a turbulence of axisymmetric state (between the pancake-like
and cigar-like turbulence). With this representation, the spacing variation leads to a changed
state of the turbulence and between the developed and developing flow conditions can be
discernible. The upstream of case $C_{3\times3}$ shows a pancake-like turbulence state. However,
the hub height and bottom tip regions shows an isotropic and axisymmetric turbulence,
respectively. The upstream of case $C_{3\times1.5}$ exhibits axisymmetric and cigar-like turbulence
in the most of the upstream domain, although the hub height region remains described by
isotropic turbulence.
Past the turbine, the four cases exhibit the turbulence of isotropic state in the hub height
region. The top tip region of the four cases shows axisymmetric turbulence although case
$C_{3\times3}$ tends to be a cigar-like turbulence. Below the hub height, the turbulence is pancake-
like and the difference amongst the cases is the covered area, where it is maximum at $C_{6\times3}$
and minimum at $C_{3\times3}$. The longest extension is found in case $C_{6\times3}$ and the lowest in case



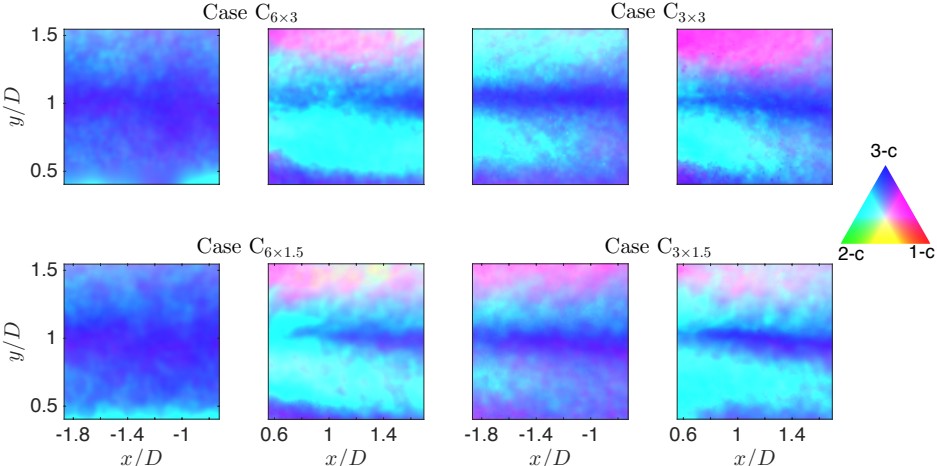

FIG. 12: Barycentric map map for the upstream and downstream of the considered cases. The small triangle is a color map key for ease of interpretation.

$C_{3\times3}$ with. Comparing to $C_{6\times3}$, the change seen in the turbulence states is starker in $C_{3\times3}$
than in $C_{6\times1.5}$, confirming that the impact of reducing streamwise spacing is greater than
changing the spanwise spacing. However, the impact of the spanwise spacing is noticeable
when $S_x$ equals $3D$. The ability to identify the turbulence structure allows for identification
of its influence on subsequent turbines in terms of fatigue loads and from the standpoint of
control, higher anisotropic character can lead to control of the flow field. It is important to
observe that the larger degree of anisotropy of the wind turbine wakes is strongly correlated
with the production of turbulent kinetic energy and mean kinetic energy entrainment and is
important to model correctly for an efficient power predicting. The stress tensor invariants,
by definition, do not depend on reflection or rotation of the coordinate system meaning that
they are unbiased descriptive for the turbulent flow.
**V. POWER MEASUREMENTS.**
Figure 12 demonstrates the power produced by each turbine, $\mathcal{F}_x$, obtained with the torque
sensing system, versus the angular velocity, $\omega$. The power measurements are normalized by
the maximum theoretical power $\frac{1}{2}\rho A_c U_\infty^3$, where $\rho$ is the air density, $A_c$ is swept area of
the turbine rotor $\pi D^2/4$. The angular velocity is normalized by the $2U_\infty/D$. It is apparent
from the figure that the maximum power is extracted at the normalized angular velocity of



$15.8 \pm 1$. The maximum normalized power of 0.062 is harvested at the largest spacing, *i.e.*, case $C_{6\times3}$. Fixing the spanwise spacing and decreasing the streamwise spacing reduces the normalized power produced by 33% for $S_x = 6D$ (from case $C_{6\times3}$ to case $C_{3\times3}$) and by 22 % for $S_x = 3D$ (from case $C_{3\times1.5}$ to case $C_{6\times1.5}$). The complementary change in spacing holds the streamwise spacing constant while decreasing the spanwise spacing. In that case the normalized power produced is reduced by 20% for $S_z = 3D$ (from case $C_{6\times3}$ to case $C_{6\times1.5}$) and by 6% for $S_z = 1.5D$ (from case $C_{3\times3}$ to case $C_{3\times1.5}$). Nilsson et al. (2015) has complementary results to the ones present, where an increase in power produced is attained in the largest spacing and conversely, decreased in the limited spacing case. Furthermore, increasing the spanwise distance has a less notable effect in comparison to the streamwise spacing.

The trend of the power curves follows the one observed in the averaged profiles of the streamwise velocity, see figure 6 (a). Further, they verify the relationship between the power of the turbine with the deficit velocity. The maximum power and velocity are found in the case $C_{6\times3}$ and the minimum quantities are noticed in $C_{3\times1.5}$. The smallest variations in the power measurement and main velocity are observed between cases $C_{3\times3}$ and $C_{3\times1.5}$, whereas the largest difference is observed between cases $C_{6\times3}$ and $C_{3\times3}$. Increased longitudinal spacing produces larger energy content in the first few modes as to provide the imprint of the flow; thus, this is reflected in an increase in power as directly measured via a torque sensing device.

## VI.  CONCLUSIONS

Insight into the behavior of the flow in a wind turbine array is useful in determining how to highlight the overall power extraction with the variation in spacing between the turbines. The work above quantifies effects of tightly spaced wind turbine configurations on the flow behavior. The findings of this study have a number of important implications, especially regarding the cost of a wind farm or when large areas are not available. Stereographic PIV data are used to assess characteristic quantities of the flow field in a wind turbine array with varied streamwise and spanwise spacing. Four cases of different streamwise and spanwise spacings are examined; the streamwise spacing being $6D$ and $3D$, and spanwise spacing being $3D$ and $1.5D$. The flow fields are analyzed and compared statistically and by





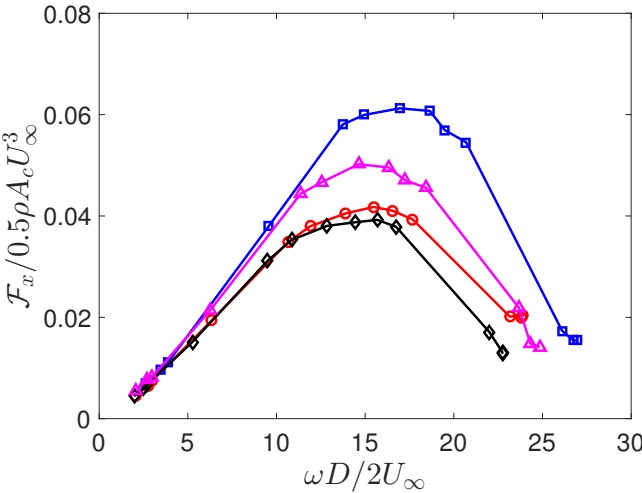

FIG. 13: Extracted power of the wind turbine at different angular velocities for four different cases $C_{6\times3}$ ($\square$), $C_{3\times3}$ ($\bigcirc$), $C_{3\times1.5}$ ($\lozenge$), and $C_{6\times1.5}$ ($\triangle$).

snapshot proper orthogonal decomposition.

390       The streamwise mean velocity, and Reynolds shear stress are quantified upstream and

downstream of the wind turbine in the considered cases. In the inflow measurement window,
higher velocities are observed in cases $C_{6\times3}$ and $C_{6\times1.5}$ comparing to the other two cases
whose inflows are unrecovered wakes from preceding rows. In contrast, case $C_{3\times3}$ and $C_{3\times1.5}$
show higher Reynolds shear stress. The notable differences between the cases are found
above the top tip and below the bottom tip downstream the turbines, whereas the core
of the wakes shows fewer discrepancies. The streamwise and spanwise spacings have a
concerted effect on the flow, where the degree of the impact of one change highly depends
on the other. This relationship is shown in all statistical quantities discussed here, such as
reducing of the streamwise spacing by 50% leads to increases in the averaged Reynolds shear
stress by 16% when $S_z = 3D$. According to current statistical quantities, one can infer that
the higher influence of streamwise spacing is shown when the spanwise spacing is $S_z = 3D$,
and the significant effect of the spanwise spacing is observed when the streamwise spacing is
$S_x = 3D$. To make comparisons independent of the effects streamwise spacing, streamwise
average profiles of the statistical quantities are computed. Averaged profiles of the velocity
follow the order of higher velocity seen in the contour plots in case $C_{6\times3}$ and lowest velocity
in case $C_{3\times1.5}$. The maximum and minimum difference are observed between cases $C_{6\times3}$ with





case $C_{3\times1.5}$ and $C_{3\times3}$ with case $C_{3\times1.5}$. The result also reveals that the streamwise spacing is more impactful than the spanwise spacing. Spatially-averaged profile of Reynolds shear stress shows the maximum and minimum values occur in cases $C_{3\times3}$ and $C_{6\times1.5}$, respectively.

Based on the POD analysis, the upstream measurement plane of the four cases converges faster than the downstream window. Case $C_{6\times3}$ and $C_{6\times1.5}$ show the rapid convergence in cumulative energy content upstream of the turbine, but $C_{6\times3}$ remains behind case $C_{6\times1.5}$ in the wake. The first mode of the case $C_{6\times1.5}$ carries the maximum turbulent kinetic energy content compared to the first mode of the other cases. No significant difference in energy content is observed after mode 10 between the four cases. The streamwise-averaged profiles of the Reynolds shear stress are reconstructed by back-projecting coefficients onto the set of eigenfunctions. Low modes are used individually to demonstrate their contributions to the overall flow. Cases $C_{6\times1.5}$ and $C_{6\times3}$ converge to the total spatially-averaged profile faster than other two cases and the discrepancy in reconstruction is mainly observed in profiles using only the first five modes. The same trend in reconstruction is observed in cases $C_{3\times3}$ and $C_{3\times1.5}$. Reconstructed profiles display the effects of the spacing, where the array of large streamwise spacing exceeds and reconstruct faster than the other cases due to carrying more coherent structure within the flow.

Based on the anisotropy stress tensor and color map visualization, the spacing modifies the turbulence structure and the longest spacing attenuates the perturbation of the turbulence, inducing the flow towards a more isotropic state. The hub height region shows an isotropic turbulence state regardless the spacing. The differences of the color map visualization between the downstream locations of the four cases show some structural dependency on the spacing between turbine rotors.

Power production by the turbines is measured directly using torque sensing system. The power curves follow the same trend as the velocity profiles. The maximum power extracted at the normalized angular velocity of $15.8 \pm 1$ and it is harvested in case $C_{6\times3}$. The small difference in harvested power is observed between cases $C_{3\times3}$ and $C_{3\times1.5}$. The current work demonstrates that wake statistics and power produced by a wind turbine depend more on streamwise spacing than spanwise spacing. However, results above pertain only to a fixed inflow direction. In the case where the bulk flow orientation changes, spacing in both the streamwise and spanwise directions will be important to the optimal power production in a wind turbine array. Continued efforts are required to understand the impact of stream-




wise and spanwise spacing in infinite array flow with Coriolis forcing and under different
stratification conditions.

### Acknowledgments

The authors are grateful to NSF-ECCS-1032647 for funding this research.

---

R. J. Barthelmie, S. T. Frandsen, M. Nielsen, S. Pryor, P.-E. Rethore, and H. Jørgensen, Wind
Energy **10**, 517 (2007).

L. P. Chamorro and F. Porté-Agel, Boundary-layer meteorology **132**, 129 (2009).

R. B. Cal, J. Lebrón, L. Castillo, H. S. Kang, and C. Meneveau, Journal of Renewable and
Sustainable Energy **2**, 013106 (2010).

M. Calaf, C. Meneveau, and J. Meyers, Physics of Fluids **22**, 015110 (2010).

L. P. Chamorro and F. Porté-Agel, Energies **4**, 1916 (2011).

R. J. Barthelmie, S. Pryor, S. T. Frandsen, K. S. Hansen, J. Schepers, K. Rados, W. Schlez,
A. Neubert, L. Jensen, and S. Neckelmann, Journal of Atmospheric and Oceanic Technology **27**,
1302 (2010).

R. J. Barthelmie and L. Jensen, Wind Energy **13**, 573 (2010).

K. S. Hansen, R. J. Barthelmie, L. E. Jensen, and A. Sommer, Wind Energy **15**, 183 (2012).

F. González-Longatt, P. Wall, and V. Terzija, Renewable Energy **39**, 329 (2012).

J. Meyers and C. Meneveau, Wind Energy **15**, 305 (2012).

R. J. Stevens, Wind Energy p. 10.1002/we.1857 (2015).

K. Nilsson, S. Ivanell, K. S. Hansen, R. Mikkelsen, J. N. Sørensen, S.-P. Breton, and D. Henningson,
Wind Energy **18**, 449 (2015).

J. Meyers and C. Meneveau, AIAA **827**, 2010 (2010).

X. Yang, S. Kang, and F. Sotiropoulos, Physics of Fluids (1994-present) **24**, 115107 (2012).

Y.-T. Wu and F. Porté-Agel, Boundary-Layer Meteorology **146**, 181 (2013).

C. L. Archer, S. Mirzaeisefat, and S. Lee, Geophysical Research Letters **40**, 4963 (2013).

R. J. Stevens, D. F. Gayme, and C. Meneveau, Wind Energy **19**, 359 (2016).

J. L. Lumley, Atmospheric Turbulence and Radio Wave Propagation pp. 166–178 (1967).




L. Sirovich, Quarterly of Applied Mathematics **45**, 561 (1987).
M. N. Glauser and W. K. George, in *Advances in Turbulence* (Springer, 1987), pp. 357–366.
P. Moin and R. D. Moser, Journal of Fluid Mechanics **200**, 471 (1989).
S. Shah and E. Bou-Zeid, Boundary-Layer Meteorology **153**, 355 (2014).
M. Tutkun, P. V. Johansson, and W. K. George, AIAA **46**, 1118 (2008).
S. Tirunagari, V. Vuorinen, O. Kaario, and M. Larmi, CSI Journal of Computing **1**, 20 (2012).
S. J. Andersen, J. N. Sørensen, and R. Mikkelsen, Journal of Turbulence **14**, 1 (2013).
N. Hamilton, M. Tutkun, and R. B. Cal, Wind Energy **18**, 297 (2015).
D. Bastine, B. Witha, M. Wächter, and J. Peinke, Journal of Physics: Conference Series **524**,
475  012153 (2014).

C. VerHulst and C. Meneveau, Physics of Fluids **26**, 025113 (2014).
N. Ali, H. F. Kadum, and R. B. Cal, Journal of Renewable and Sustainable Energy **8**, 063306
478  (2016).

J. Rotta, Z. Physik **131** (1951).
S. B. Pope, *Turbulent flows* (Cambridge University Press, 2000).
J. L. Lumley and G. R. Newman, Journal of Fluid Mechanics. **82**, 161 (1977).
S. Banerjee, R. Krahl, F. Durst, and C. Zenger, Journal of Turbulence **8**, N32 (2007).
M. Emory and G. Iaccarino, Annual Brief, Center for Turbulence Research (2014).
R. A. Antonia, J. Kim, and L. Browne, Journal of Fluid Mechanics. **233**, 369 (1991).
P. Krogstad and L. E. Torbergsen, Flow, turbulence and combustion. **64**, 161 (2000).
R. Gómez-Elvira, A. Crespo, E. Migoya, F. Manuel, and J. Hernández, Journal of Wind Engineering
and Industrial Aerodynamics. **93**, 797 (2005).
N. Hamilton and R. B. Cal, Physics of Fluids. **27**, 015102 (2015).
H. S. Kang and C. Meneveau, Measurement Science and Technology **21**, 105206 (2010).