# Peer review of "Assessing Spacing Impact on Coherent Features in a Wind Turbine Array Boundary Layer"

_Wind Energy Science, 2017_

## Referee Comment (RC1) · Anonymous Referee #1 · 6 Sep 2017

Review of the manuscript "Assessing spacing impact on coherent features in a wind turbine array boundary layer" by Naseem et al.

Overview:

This manuscript presents results of an experimental wind tunnel study of four different spacing configurations of a 4 x 3 wind turbine array. The authors investigated the mean and turbulent features of the flow using techniques such as the snapshot POD and anisotropy stress tensor. The manuscript is overall well written, but some major clarifications are needed before it can be published in Wind Energy Science. My main concern about the manuscript is the lack of physical interpretation of POD analysis. Please see my specific comments below.

Specific comments:

1. You should mention in the literature review part in Introduction that the spacing between wind turbines and their layout are also function of orography among other things, and not just wind direction. For example, see the following recent reference and references therein:

Romanic D, Parvu D, Refan M, Hangan H. 2018. Wind and tornado climatologies and wind resource modelling for a modern development situated in "Tornado Alley." *Renewable Energy* **115**: 97–112. DOI: 10.1016/j.renene.2017.08.026.

Then you can say that your paper, however, is restricted to a flat surface and topographic influences are not considered.

2. In Eqs. (1) and (2), please explain all variables although some of them might be trivial. What are $N$, $A$, $\Omega$? In Line 94, you are explain $R$ that does not seem to appear in the previous equations.

3. Line 102. The sum in the nominator in equation for $\eta_n$ shouldn't go to the same value as the index $n$. Please explain these formulas accordingly.

4. What was the blocking ratio for your wind tunnel tests?

5. Line 159. You are using a closed-circuit wind tunnel where the flow is mechanically generated. What do you mean by neutral stratification? You haven't checked for atmospheric stability or at least I don't see any stability parameters in your paper (e.g. potential temperature profiles, Richardson number, etc.).

6. What are the geometric and velocity scales (and thus time scales) in your experiments? You did provide the geometric details of your wind turbine models, but what full-scale wind turbine (or turbines) are you replicating in your wind tunnel experiments?

7. Why is there uncertainty of $U_\infty$ in Line 183? That is, why the value is "about 5.5 m s$^{-1}$?" Please provide additional explanation.

8. Figure 11. You use the same color in the insets to represent three different ranges of models so the reader needs to guess which line represents which range. Please introduce either additional colors or use symbols or dashed lines.

9. Lines 256–266. You are implying that the 3% difference in the turbulent kinetic energy is large whereas the differences between the cases $C_{6x3}$ and $C_{6x1.5}$ are small. Since you didn't quantify the differences

between the cases $C_{6x3}$ and $C_{6x1.5}$, I would argue that the 3% difference is the turbulent kinetic energy is also small.

10. It is not clear to me from the sentence in Line 256 why did you choose to show only the first, the fifth and the twentieth modes? Why not for example the second mode or eighteenth or any other modes? Please clarify in more details.

11. Are there any physical meanings behind the modes that you showed? What flow physics they show if any? It is very important to relate the pure mathematics of POD with the flow physics. That being said, please provide some physical explanations of the modes. Please note that this comment must be addressed seriously before I give a positive recommendation to this manuscript.

12. It is typical in the field of fluid dynamics and turbulence to use the terms such as hairpin turbulence instead of cigar-like turbulence. This comment however is just a suggestion so you can keep cigar-like terms if you prefer it.

13. Section IV E (Anisotropy Stress Tensor). What are you trying to show in this section that would be of importance in wind energy industry? That is, what are the practical applications of your results? In Line 346 you mentioned that it can have implications in the terms of fatigue loads, but the statement is too general. Please provide more explanations and some references would also be very good.

Grammar:

1. Line 9. "an increase" instead of "an increased"

2. Lines 15–16. "in turn is a function of the surface roughness"

3. Line 22. The reference should be in brackets not an in-line format.

4. If you want to use Roman numbers to denote chapters then remove dots after the numbers.

5. Line 140. "closed-circuit" instead of "closed- circuit'

6. When you write $ms^{-1}$, please have a space between m and s because without the space it looks like millisecond (an example is in Line 183). The same rule applies to other units.

7. Here I provided just some of the grammatical mistakes that I found. I advise the authors to proofread the manuscript few more times.

---

## Author Comment (AC1) · 20 Sep 2017

The comment was uploaded in the form of a supplement:
https://www.wind-energ-sci-discuss.net/wes-2017-32/wes-2017-32-AC1-supplement.zip

---

## Referee Comment (RC2) · Anonymous Referee #2 · 9 Oct 2017

The paper applies POD and Barycentric color map to analyze the wake field in an array of wind turbines with different spacing. The paper is generally well-written (though some grammatical errors do exist). The POD analysis provides insights into the dominant structures in the wake field, which ultimately will be valuable for finding reduced order models for the wakes. The barycentric map highlights the specific anisotropic features in the wakes, which again will be useful for gauging models for the wakes. The analysis is sound and reasonably complete. I recommend publication with minor corrections. Some specific comments are listed below:

Line 89: From this line, am I correct to understand that the POD is applied to the fluctuating velocity only, meaning the mean velocity is subtracted first? Some clarifications are needed.

Page 5: Eq (2) contains typo. In the next sentence (line 94) R(x,x') was referred to as if it appeared in Eq. (2). However it is not there. Also, the locations x and x' should have an arrow on the top to be consistent.

Line 97: it would be more helpful to explain the relation between G and the coefficients A in Eq. (1).

Line 102: the running index in the summation on the top should be different from n, as n is the upper bound of the summation.

Line 108: Please give the definition of k.

Line 115: 'to additional promote the study of...'? It does not quite make sense.

Line 129: I suspect that the $C^*_{ic}$ inside the parentheses should not have the asterisk. Also, the coefficients 5 and 0.65 are different from those in Emory and Iaccarino. Some explanation is needed.

Line 162: 'represent aligned wind farms' and 'is not considered'.

Line 173: the typesetting of the expression 0.2m x 0.2m is a bit awkward, thought I suppose this can be fixed by the publisher.

Line 239: 'reduction of the streamwise spacing'

Fig. 8: It seems that, apart from case C3X3, the upstream structure is very much different from the downstream one. Can the authors please comment? This part represents a main contribution of the article. It is important to give an in-depth analysis.

Line 279: Should be 'the intermediate modes are associated with the inflow characterizations'?

Line 354: Fig 12 should be Fig 13

Line 422: '...streamwise spacing exceeds and reconstruct faster...'? Something is missing here. It does not quite make sense.

---

## Author Comment (AC2) · 11 Oct 2017

The comment was uploaded in the form of a supplement:
https://www.wind-energ-sci-discuss.net/wes-2017-32/wes-2017-32-AC2-supplement.zip

———————————————

---

## Referee Comment (RC3) · Anonymous Referee #3 · 25 Nov 2017

The authors presented an interesting study in particular in revealing new results that have not been published earlier. The manuscript is well designed in structure and clearly presented. To this reviewer, the manuscript is worth being published. However, there are minor issues need to be further clarified or revised. Below, the issues are presented.

- 1. Page 3, line 22: For the case of Nysted farm, the wind direction and mean wind direction are given. It would be useful to include the reason of importance of this information.
- 2. Page 8, line 142: It would be very informative to compute the blockage ratio based on the cross-section area of the tunnel and the turbines' area in order to assess if the ratio is in the acceptable range.
- 3. Page 8, line 146: It is stated that acrylic strakes were used to modify the upstream inflow. To this reviewer, it would be helpful to provide information about the velocity distribution upstream the turbines to depict the formation of the boundary layer (B.L.). Furthermore, the relation of the B.L. to the turbines can be compared to the realistic situation.
- 4. Page 9, line 150: Based on the provided information, the turbines are miniature ones. To this reviewer, it is necessary to include the assessment of the scaling effects in particular the effect of Reynolds number.
- 5. Page 9, line 150: It is mentioned in the manuscript that 3-bladed horizontal axis wind turbines are used in the experiments. Is there any specific design used to construct these turbines? Or they are scaled-down versions of an existing design?
- 6. Page 9, line 166: It would be informative to include the sampling rate of the SPIV system.
- 7. Page 9, line 176: To this reviewer it would useful to provide error calculation for the PIV measurements.
- 8. Page 10, FIG 3: According to this figure, the turbines are located close to the lateral walls? How close is this distance? And how much effect was noticed in the measurements?
- 9. Page 16, FIG 8, 9 and 10: It would be helpful to include legends for these figures in order to compare different modes.

As a general recommendation, to this viewer, it is necessary to provide reasonable physical interpretations for curves, profiles and contour maps presented in the manuscript. In other words, it would be more informative to present physical reasons associated with the curves' behaviours. This can be taken into account for example in page 6- FIG 6 (e.g. line 222: "a reduction in streamwise spacing shows less effect when the spanwise spacing Sz= 1.5D." what is the physical reason?), and page 18, FIG 11.

---

## Author Comment (AC3) · 30 Nov 2017

The comment was uploaded in the form of a supplement:
https://www.wind-energ-sci-discuss.net/wes-2017-32/wes-2017-32-AC3-supplement.zip

———————————————

---

## Author Response (AR1)

The authors thank the editor for his effort in reviewing the paper and making valuable comments about the work. We have revised the manuscript and considered all suggestions. As a result, the paper has been significantly strengthened. Point-by-point answers to the editor's comments are provided below:

**• A.1: Please make sure that all suggestions are implemented.**

**Response A.1:** Thanks. The authors have been addressed all comments and implemented all the suggestion from the reviewers. The highlighted version of the manuscript contain these points in **bold** font.

• A.2: We consider that after the review the paper has improved substantially mostly in terms of a more physical interpretation of the results both from flow and implications perspective.

**Response A.2:** The authors confirm that physical interpretation and implications of the results have been included as suggested by the reviewers. The added comments have been highlighted in the manuscript in bold and are also provided herein for completeness of this document.

The normalized mean streamwise velocity and the turbulence intensity in Nilsson et al. (2015) showed similar compound wakes from the upstream and downstream turbines and confirmed the current result of cases  $C_{3\times3}$  and  $C_{3\times1.5}$ . In that study, there was one location with an absent turbine and the flow was given extra space for recovery. The recovered wake flow in Nilsson et al. (2015) is similar to the present cases  $C_{6\times3}$  and  $C_{6\times1.5}$ .

In the downstream window, comparison indicates that reducing streamwise spacing increases the Reynolds shear stress.

...confirming that the impact of reducing streamwise spacing is greater than changing the spanwise spacing. Interestingly, when the spanwise spacing is fixed to  $S_z = 1.5D$ , changing the streamwise spacing has a smaller than expected effect. Constraining the wake suppresses development of the mean velocity in the streamwise and spanwise directions.

However, the upstream and downstream windows of case  $C_{3\times 1.5}$  are more similar in terms of turbulence and organization. From mode 2 through 10, the starkest difference between the upstream and downstream is found in case  $C_{6\times 3}$ . Increasing the characteristic area per turbine provides room for the flow to become more homogeneous in the upstream window and exhibit the most significant momentum deficit in the wake, accounting for the differences seen in  $\eta_n$  upstream and downstream.

Although, after mode 10, there is no significant difference in the energy content from case to case, the structure of the modes shows a significant discrepancy between the cases confirming that the intermediate modes are associated with the inflow characterizations. Thus, the intermediate modes are responsible for carrying the significant part of flow dynamic and cooperative behavior in the energy cascade. Therefore, any low-order models should include these intermediate modes in order to improve the behavior dramatically and capture the dynamic of the full system.

In cases  $C_{3\times3}$  and  $C_{3\times1.5}$ , less energetic features arise from the reduced spacing effect that leads to a reduction of the mean velocities within the canopy and an increase in lateral wake interactions. These interactions, which become larger as a result of the accumulated wakes, expand downstream of the rotor. Thus, the streamwise spacing allows for the flow to recover and therefore produce larger, more coherent structures within the domain, which in comparison eclipses variations produced by the spanwise spacing. Also, the large spacing offers a larger frontal area to the wind coming from above the lateral sides. The ability to identify the turbulence structure allows for identification of its influence on subsequent turbines in terms of fatigue loads (Frandsen and Thøgersen 1999). Further, regions of the flow that are characterized by highly anisotropic turbulence are those in which one is likely to find large-scale, coherent turbulence structures. These structures impart the greatest axial and bending loads onto subsequent turbine rotors leading to accelerated fatigue and increased operational and maintenance costs for wind farms. In addition, regions of high anisotropy correlate with gradients in the mean flow and turbulence (Hamilton and Cal 2015). These quantities are of particular interest in wind farm modeling and design. Accordingly, the accurate representation of gradients in wind farm design modeling is a necessary check to accurately representing production of and flux by turbulence kinetic energy, wake interaction, and structural loading on constituent turbines. Finally, the stress tensor invariants, by definition, do not depend on reflection or rotation of the coordinate system meaning that they are unbiased descriptors for the turbulent flow (Pope 2000).

In closing, we thank the editor again for the useful feedback and thorough review of the manuscript.

[revised manuscript text omitted]